# Differentiated care for youth in Zimbabwe: Outcomes across the HIV care cascade

**Chido Dziva Chikwari**[1,2]ꙮ*, **Katharina Kranzer**[1,3,4]ꙮ, **Victoria Simms**[1,2], **Amani Patel**[2], **Mandikudza Tembo**[1,2], **Owen Mugurungi**[5], **Edwin Sibanda**[6], **Onismo Mufare**[1], **Lilian Ndlovu**[1], **Joice Muzangwa**[1], **Rumbidzayi Vundla**[1], **Abigail Chibaya**[1], **Richard Hayes**[2], **Constance Mackworth-Young**[1,7], **Sarah Bernays**[8], **Constancia Mavodza**[1,2], **Fadzanayi Hove**[1], **Tsitsi Bandason**[1], **Ethel Dauya**[1], **Rashida Abbas Ferrand**[1,3]

1 The Health Research Unit Zimbabwe, Biomedical Research and Training Institute, Harare, Zimbabwe, 2 MRC International Statistics and Epidemiology Group, Department of Infectious Disease Epidemiology, London School of Hygiene and Tropical Medicine, London, United Kingdom, 3 Department of Clinical Research, London School of Hygiene and Tropical Medicine, London, United Kingdom, 4 Department of Infectious Diseases and Tropical Medicine, Ludwig-Maximilians-Universität, Munich, Germany, 5 AIDS and TB Unit, Ministry of Health and Child Care, Harare, Zimbabwe, 6 City Health Department, Bulawayo City Council, Bulawayo, Zimbabwe, 7 Department of Global Health and Development, London School of Hygiene and Tropical Medicine, London, United Kingdom, 8 Faculty of Medicine and Health, University of Sydney, Sydney, Australia

ꙮ These authors contributed equally to this work.
* chido.dzivachikwari@lshtm.ac.uk

**Data Availability Statement:** Individual, anonymised participant data and a data dictionary is available through the London School of Hygiene & Tropical Medicine repository (Data Compass). It

## Abstract

Youth living with HIV are at higher risk than adults of disengaging from HIV care. Differentiated models of care such as community delivery of antiretroviral therapy (ART) may improve treatment outcomes. We investigated outcomes across the HIV cascade among youth accessing HIV services in a community-based setting. This study was nested in a cluster-randomised controlled trial (CHIEDZA: Clinicaltrials.gov, Registration Number: NCT03719521) conducted in three provinces in Zimbabwe and aimed to investigate the impact of a youth-friendly community-based package of HIV services, integrated with sexual and reproductive health services for youth (16–24 years), on population-level HIV viral load (VL). HIV services included HIV testing, ART initiation and continuous care, VL testing, and adherence support. Overall 377 clients were newly diagnosed with HIV at CHIEDZA, and linkage to HIV care was confirmed for 265 (70.7%, 234 accessed care at CHIEDZA and 31 with other providers); of these 250 (94.3%) started ART. Among those starting ART at CHIEDZA who did not transfer out and had enough follow up time (>6 months), 38% (68/177) were lost-to-follow-up within six months. Viral suppression (HIV Viral Load <1000 copies/ml) among those who had a test at 6 months was 90% (96/107). In addition 1162 clients previously diagnosed with HIV accessed CHIEDZA; 714 (61.4%) had a VL test, of whom 565 (79.1%) were virally suppressed. This study shows that provision of differentiated services for youth in the community is feasible. Linkage to care and retention during the initial months of ART was the main challenge and needs concerted attention to achieve the ambitious 95-95-95 UNAIDS targets.

can be accessed from this DOI: https://doi.org/10.17037/DATA.00003649

**Funding:** The CHIEDZA study is funded by the Wellcome Trust (Senior Fellowship to RAF: 206316/Z/17/Z). VS and RH were partially supported by the UK Medical Research Council (MRC) and the UK Department for International Development (DFID) under the MRC/DFID Concordat agreement which is also part of the EDCTP2 programme supported by the European Union Grant Ref: MR/R010161/1.The funders did not contribute to the study design, data collection and analysis, decision to publish nor preparation of the manuscript

**Competing interests:** I have read the journal's policy and the authors of this manuscript have the following competing interests: Chido Dziva Chikwari has been a guest Editor for Plos Global Public Health. The other authors have declared that no competing interests exist.

# Introduction

In 2019, an estimated 3.9 million young people (aged 15–24 years) were living with HIV globally, the majority in Southern Africa [1]. Young people continue to be the age group with the highest HIV incidence, especially among women in whom HIV incidence is three times higher than in males of the same age [2].

In 2014, UNAIDS set the 90−90−90 target aiming for 90% of people living with HIV to be aware of their HIV status, 90% of people diagnosed with HIV to be taking antiretroviral therapy (ART), and 90% of those on ART to have a suppressed HIV viral load, by 2020 [3]. These targets were followed by the even more ambitious 95-95-95 targets, to be achieved by 2025 [4]. While Southern African countries have made great progress towards achieving these targets, progress has been much slower for young people living with HIV in the region [5−10].

Young people face challenges across each step of the HIV care cascade beginning with knowing their HIV status through testing, timely linkage to and continued engagement with HIV care, and optimal care outcomes including viral suppression [11−16]. Successful interventions to increase coverage of HIV testing among young people have included community-based HIV testing including HIV self-testing, and HIV testing through outreach programmes at educational facilities and at fixed community locations [17−21].

Once diagnosed linkage to and continued engagement with care is essential to achieve sustained viral suppression [17]. Young people are more likely than any other age group to disengage [22]. Barriers to retention include financial dependency, stigma, health services that are not youth friendly and clinic opening times that are incompatible with school attendance, and challenges with adaptations for care as they transition to adulthood [23−25].

Simplified and patient-centred care options tailored to each stage of the patient care journey ('differentiated models of service delivery') aimed at improving engagement and retention in care have been developed for, and evaluated in, adults living with HIV. However, there are scarce data on the implementation and effectiveness of these interventions among young people [26].

The aim of this paper is to investigate the outcomes at each step of the HIV care cascade in youth accessing a differentiated model of service delivery offering community-based integrated HIV and sexual reproductive health services for young people aged 16–24 years in three provinces in Zimbabwe.

# Methods

## Study design, intervention and setting

This study was nested within a cluster randomised trial (CHIEDZA) that aimed to investigate the impact of a comprehensive community-based package of HIV services, integrated with sexual and reproductive health services and general health counselling for youth aged 16–24 years, on population-level HIV viral suppression. The trial was conducted in three urban and peri-urban provinces across Zimbabwe, each with eight clusters randomised 4:4 to control (existing health services) or intervention clusters [27]. A cluster was defined as a geographically demarcated area with a multi-purpose community centre and a primary care clinic (PCC). In intervention clusters, integrated HIV and sexual and reproductive health services were delivered weekly from this centre to cluster residents aged 16–24 years. Services included HIV testing, and HIV treatment and adherence support for those living with HIV through support groups and counselling, as well as sexual and reproductive health services (offered to all eligible clients regardless of HIV status): these included contraception, menstrual hygiene management, syndromic management of sexually transmitted infections, risk reduction and general

health counselling. Services were offered by a multidisciplinary team of service providers over 30 months [27].

The intervention, co-designed with youth, was specifically configured to be "youth friendly", and the intervention team was selected based on prior experience of working in communities and with youth [28]. A structured training programme included practical training on each of the intervention components as well as training on provision of youth friendly services, particularly communication and counselling that is appropriate to age and maturity, sexual orientation and attitudinal training specifically emphasizing respect, confidentiality, non-judgement, and relatability. Debrief meetings with the intervention teams were held two-weekly to monthly and incorporated problem-solving, discussion of complex cases and operational issues to ensure that intervention providers were supervised and mentored.

The start of the intervention period was staggered, with Harare province starting on 1 April 2019, followed three and six months later in Bulawayo and Mashonaland East provinces respectively. The intervention period ended on 31 March 2022. All services were voluntary and offered free of cost.

## HIV testing and care services

HIV testing was conducted according to national guidelines [29]. Those who tested HIV-positive were offered a choice of being referred to the PCC in the cluster (with the client accompanied to the PCC by a CHIEDZA provider to help facilitate linkage to care) or of accessing care in the community through CHIEDZA. If the latter was selected, the young person was assigned a national HIV programme number and their HIV records were maintained at the PCC. For those who accessed care through CHIEDZA, ART was supplied by the PCC through the national HIV programme. CHIEDZA staff updated the PCC data when they collected ART for supply through CHIEDZA. This ensured that young people remained part of the national HIV programme allowing them to transition to receiving care from any PCC of their choice at the end of the intervention period. Clients living with HIV, whether newly diagnosed or previously diagnosed, who opted to receive HIV care at CHIEDZA were referred to as the CHIEDZA HIV cohort. Clients living with HIV and already accessing care outside CHIEDZA were encouraged to remain with their routine care provider. However, those who wished to transfer care to CHIEDZA were able to do so.

HIV treatment was provided according to national guidelines [30] and there was a defined referral pathway to a health facility for any clinical indications (e.g., severe drug toxicity or incident symptoms, suspected treatment failure). A point-of-care CD4 count was done at time of diagnosis or ART initiation at the CHIEDZA site. At the first consult following HIV diagnosis, a CD4 count was performed: those with a CD4 count <100 cells/uL had a serum cryptococcal antigen test. Assessment also included a WHO tuberculosis symptom screen, with sputum Xpert MTB/Rif testing, chest radiography (and/or investigations for extrapulmonary tuberculosis as relevant) performed offsite in those who screened positive. Following ART initiation, clients were followed up and accessed ART through the CHIEDZA centres.

Young people living with HIV (regardless of whether they accessed HIV care through CHIEDZA or not) were offered HIV viral load (VL) testing after six months on ART. Clients in the CHIEDZA HIV cohort were ineligible for VL testing if they transferred elsewhere before completing six months on ART at CHIEDZA, or if the CHIEDZA intervention ended they completed six months on ART. In addition, HIV VL testing for clients seeking care outside CHIEDZA was only introduced after September 2019. VL testing was performed offsite on GeneXpert assay (Cepheid, South Africa) with results available within approximately one week. Enhanced adherence counselling was provided at CHIEDZA if not virally supressed

(>1000 copies/ml), as per Zimbabwe national guidelines. Those who did not link to care or missed scheduled appointments either before or after ART initiation were contacted by telephone, and if contact was not made, home visits (on those who had provided consent at enrolment) were undertaken.

All CHIEDZA clients living with HIV (regardless of where they accessed HIV care) were invited to join a CHIEDZA Adolescent Peer Support (CAPS) group that was modelled on the existing Community ART Refill Groups (CARGs) implemented in Zimbabwe [31]. CAPS groups were held approximately quarterly and were facilitated by the CHIEDZA nurse, counsellor and a youth worker. CAPS included a discussion session on an issue chosen by attendees and social activities, as well as the opportunity for ART refill and individualised counselling.

## Data management and statistical analysis

Demographic information, (sex, date of birth) and services received were collected on an electronic tablet for each CHIEDZA client and linked to an individual ID that was generated by software that converted a finger print into a unique ID (SIMPRINTS, UK). For CHIEDZA HIV cohort clients, an additional HIV Cohort ID number was generated and used to record HIV-specific clinical information on an electronic case report form (CRF) completed by the research team. Separate paper CRFs captured VL and CD4 test results.

Data was analysed using Stata 17.0 (StataCorp, USA). The research team followed up clients to ascertain whether they had linked to care and initiated ART elsewhere, or whether they had relocated, refused to respond, were not ready to initiate ART, or had been referred for another condition.

Clients were defined as previously diagnosed with HIV based on self-report, and as newly diagnosed if they self-reported negative or unknown HIV status and then tested HIV positive at CHIEDZA. Linkage to care was defined as having registered with any HIV clinic. Clients who did not link to care at CHIEDZA were followed up to determine whether they linked to care elsewhere. Clients were coded as linked to care elsewhere if they reported that they were receiving HIV care from another provider, with no time limit. Similarly, clients were coded as linked to care at CHIEDZA if they ever took up HIV care at CHIEDZA. Clients who were uncontactable were coded as having an unknown outcome and those who were contacted at least once but stopped responding thereafter were coded as lost to follow-up. While viral suppression was defined as VL<1000copes/ml as per national guidelines, additional exploratory analysis assess proportion of youth with a VL <20copies/ml, to enable comparison with studies in other settings.

## Ethics

Ethical approval for the CHIEDZA study was obtained from the Medical Research Council of Zimbabwe [MRCZ/A/2387], the Biomedical Research and Training Institute Institutional Review Board [AP149/2018] and the London School of Hygiene & Tropical Medicine Ethics Committee [16124/RR/11602]. All intervention attendees provided verbal consent for services. The requirement for guardian consent to access services for 16–18-year-olds was waived by the ethics committees.

## Role of funding source

The funders of the study had no role in study design, data collection, data analysis, data interpretation, writing of the report, or the decision to submit the study for publication.

## Results

A total of 377 clients were newly diagnosed with HIV at CHIEDZA, while an additional 1162 clients living with HIV accessed the CHIEDZA service during the intervention (April 2019-May 2022) (Fig 1). Table 1 describes the baseline characteristics of clients newly diagnosed with HIV and those with a known HIV positive status. The majority of clients identified with HIV were women (newly diagnosed 336/377, 89.1% and previously diagnosed 1035/1156, 89.5%) and in both groups the median age was 21 (IQR 19–23).

Among those newly diagnosed 265/377 (70.3%) linked to HIV care 234/265 (88.3%) at CHIEDZA and 31/265 (11.7%) with other service providers. Of the remaining 112 (29.7%), 74 did not link to care and for 38 linkage to care status was unknown (Fig 1 and Table 1). Clients aged 16–19 years newly diagnosed with HIV were less likely to link to care (66/97, 68%) when compared to those aged ≥20 years (199/242, 82.2%). Clients who did not link to care: i) said they were not ready to start ART (n = 11), ii) refused to be contacted again, but were not on ART at the last contact (n = 14) or iii) were initially contactable following the positive HIV test, but were eventually lost to follow-up after multiple contacts encouraging them to link to care (n = 49). Those whose linkage status was unknown: i) had relocated to another community (n = 4), ii) lived outside the intervention cluster (n = 5) iii) had been referred to other service providers (n = 8) or iv) had left the CHIEDZA service after having tested HIV positive without providing any contact information (n = 21).

A minority of clients newly diagnosed with HIV and linked to care did not start ART during follow-up. Of those newly diagnosed and who linked HIV care at CHIEDZA, 9/234 (3.8%)

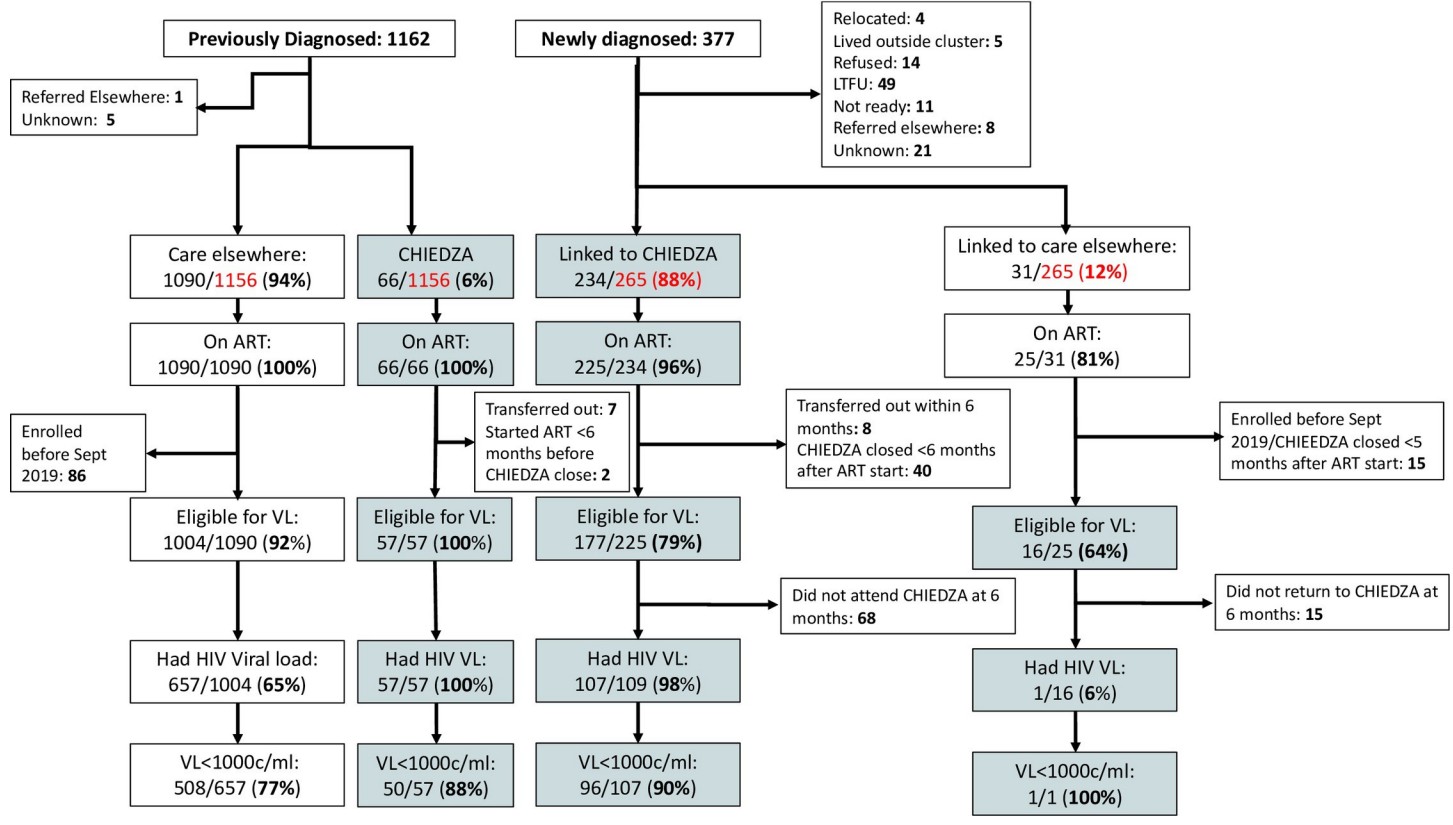

**Fig 1. Flowchart of young people living with HIV who accessed CHIEDZA services.**

**Table 1. Baseline characteristics of those newly diagnosed and previously diagnosed with HIV.**

| Characteristics | | Newly diagnosed, n (row %) | | | | | Previously diagnosed, n (row %) | | |
|---|---|---|---|---|---|---|---|---|---|
| | | Total (n = 377) | Linkage unknown (n = 38) | No linkage (n = 74) | Linked elsewhere (n = 31) | Linked to CHIEDZA (n = 234) | Total (n = 1156)* | Transferred care to CHIEDZA (n = 66) | In care elsewhere (n = 1090) |
| Age (years) | 16–19 | 114 | 17 (14.9) | 31 (27.2) | 7 (6.1) | 59 (51.8) | 328 | 22 (6.7) | 306 (93.3) |
| | 20–24 | 263 | 21 (8.0) | 43 (16.4) | 24 (9.1) | 175 (66.5) | 828 | 44 (5.3) | 784 (94.7) |
| Sex | Male | 41 | 8 (19.5) | 8 (19.5) | 1 (2.4) | 24 (58.5) | 121 | 3 (2.5) | 118 (97.5) |
| | Female | 336 | 30 (8.9) | 66 (19.6) | 30 (8.9) | 210 (62.5) | 1035 | 63 (6.1) | 972 (93.9) |
| Province | Harare | 124 | 12 (9.7) | 19 (15.3) | 12 (9.7) | 81 (65.3) | 365 | 10 (2.7) | 355 (97.3) |
| | Bulawayo | 92 | 6 (6.5) | 29 (31.5) | 3 (3.3) | 54 (58.7) | 353 | 3 (0.9) | 350 (98.3) |
| | Mashonaland East | 161 | 20 (12.4) | 26 (16.2) | 16 (9.9) | 99 (61.5) | 438 | 53 (12.1) | 385 (87.5) |
| Self-reported HIV status | Unknown | 163 | 23 (14.1) | 28 (17.2) | 13 (8.0) | 99 (60.7) | NA | NA | NA |
| | Tested HIV-ve >6mths ago | 173 | 12 (6.9) | 36 (20.8) | 16 (9.3) | 109 (63.0) | NA | NA | NA |
| | Tested HIV-ve <6mths ago | 41 | 3 (7.3) | 10 (24.4) | 2 (4.9) | 26 (63.4) | NA | NA | NA |

*Excluding 6 for whom ART treatment status is unknown

did not start ART and of those linked to other providers, 6/31 (19.4%) did not start ART. Among those who started ART in CHIEDZA, 48 either transferred out or did not have enough follow-up time (<6 months) to be eligible for a VL test. In addition, 68/177 (38.4%) did not attend follow-up visits at CHIEDZA six months after starting ART and their ART status thereafter was unknown. Among those who stayed in care with CHIEDZA >6 months after starting ART, VL suppression was 89.7% (96/107). Using <20 copies per ml as the cut off; viral suppression in this group was 77.6% (83/107).

A small proportion (66/1162, 5.7%) of clients previously diagnosed with HIV transferred their care to CHIEDZA. Transfer was more frequent in Mashonaland East 53/428 (12.4%) compared to other provinces. Among clients who underwent VL testing, viral suppression was 88% (50/57). Among eligible clients previously diagnosed with HIV and receiving care from other providers, 65.4% (657/1004) were offered and agreed to have a viral load test and 77.3% (508/657) were virally suppressed.

CD4 counts were available for 228/234 (97.4%) newly diagnosed clients linked to care in CHIEDZA: 22 (10%) had a CD4 count <200 and the median CD4 count was 496 (IQR 367–649). Of the four clients with a CD4 count <100 cells/uL , three had a serum cryptococcal antigen test, and all tests were negative. No client was diagnosed with tuberculosis.

Of 1539 clients living with HIV who accessed CHIEDZA 323 (21.0%) registered with CAPS. CAPS registration was more frequent among those accessing HIV care at CHIEDZA (110/300, 36.7%) compared to those accessing HIV care from other providers (213/1121, 19.0%). A total of 32 CAPS sessions with a total of 734 attendances were held across the three provinces (Fig 2). National and local lockdowns due to the COVID-19 pandemic severely affected CAPS groups both with regards to hosting the events and attendance.

## Discussion

We report outcomes across the HIV care cascade from a differentiated service delivery model for and co-developed with young people. The CHIEDZA intervention offered HIV testing integrated with sexual and reproductive health services, ART initiation, treatment monitoring

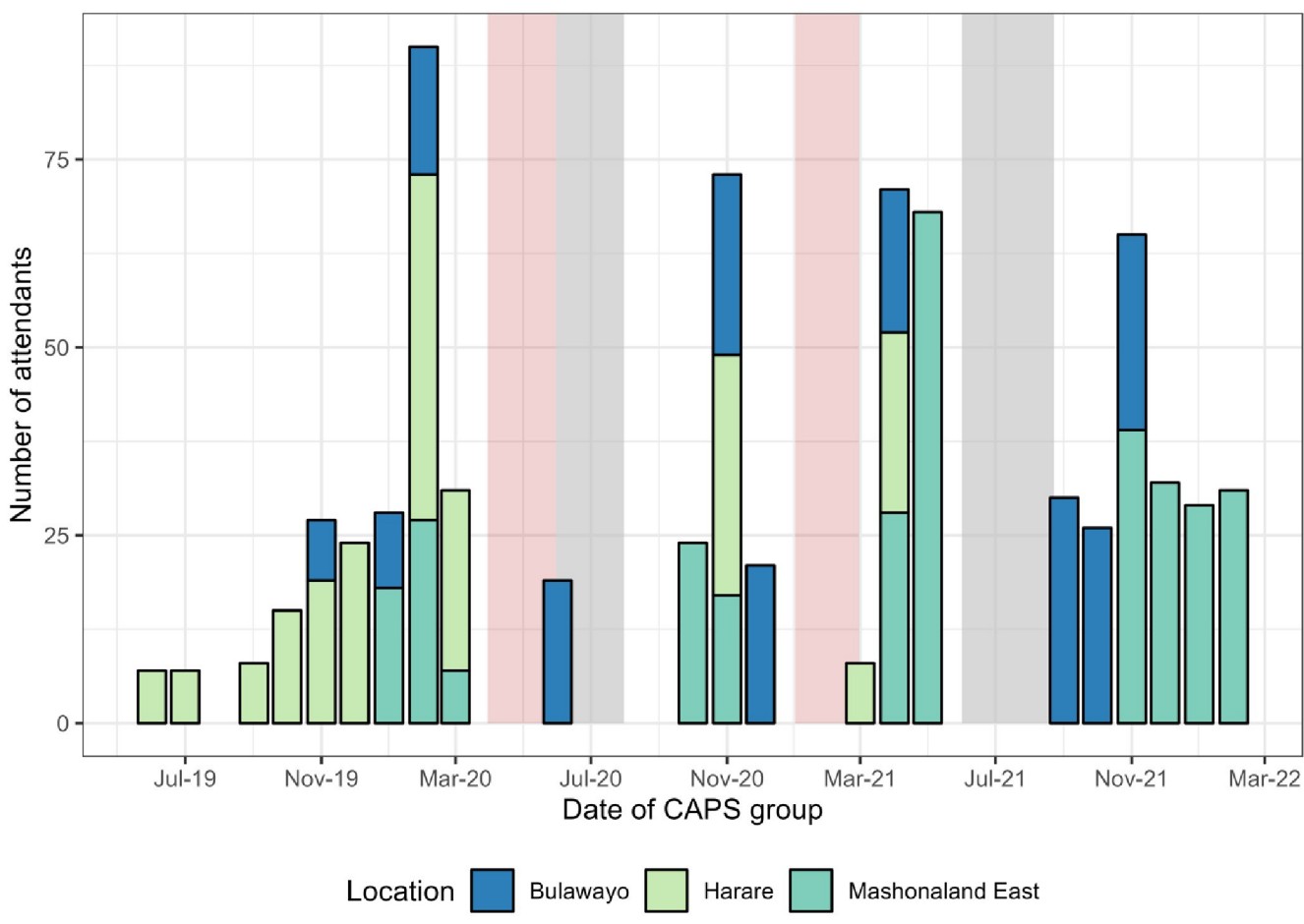

**Fig 2. Number and timing of CHIEDZA young people peer support (CAPS)).** Grey shaded areas travel restrictions and reduced working hours, Pink shaded areas indicate national lockdowns.

(HIV VL testing) and adherence support (CAPS groups) in the community. Importantly the service was not just youth friendly, but offered choice and service providers respected young people's autonomy. For example young people newly diagnosed with HIV who linked to care within CHIEDZA were provided with continuous care and services regardless of their decision on whether to start ART. Among young people who were newly diagnosed with HIV, 70.3% were confirmed to have linked to care, among those linked to care 94.3% started ART and among those starting ART at CHIEDZA and who remained in care beyond six months 89.7% were virally suppressed. However, a considerable proportion of young people either did not link to care or their linkage status was unknown. Some of them found it difficult to accept the diagnosis and either refused further contact or left the services without providing any contact information.

Importantly 38% of young people newly diagnosed with HIV disengaged from care within the first six months of having initiated ART. Early attrition from ART is common among young people and has been reported in other settings [11, 12, 32, 33]. Part of investing in retaining youth newly diagnosed with HIV in care is potentially doing further work in preparing young people for the implications of testing. This can be done by ensuring there is more in place at the point of testing to improve HIV treatment literacy and protect against individuals

disengaging after testing [11]. This could be nested within HIV pre-test counselling and incorporate messaging about viral suppression and U = U at the point of diagnosis among those newly diagnosed.

Linkage to care following HIV testing in the community is challenging. Those accessing community HIV testing services may be less likely to access clinic-based services. Additionally, community-based services may not be able to provide the same frequency (5–6 days per week) of services and/or working hours as clinic-based services. Linkage to care and ART initiation is rarely offered as part of community-based services. A systematic review including 14 studies focused on community-based HIV testing reported proportions linked-to-care ranging from 10–79% over 1–12 months of observation [34]. None of the studies included in the review focused exclusively on young people and adolescents (16–17 year olds) were generally excluded. A more recent study, "PopART for Youth" (P-ART-Y), delivered a combination HIV prevention package to 10–19 year olds in Zambia and South Africa via a door-to-door approach. The prevention package included HIV testing, supported linkage to care for those living with HIV and ongoing ART adherence support. In a before/after comparison ART coverage among adolescents living with HIV increased from 61.3% in Zambia and 65.6% in South Africa to 78.7% in Zambia and 87.8% in South Africa after one year of the intervention. While the intervention had the biggest impact on the first UNAIDS 95-95-95 target (i.e. proportion with HIV who knew their status), linkage to care must have been effective in order to increase ART coverage overall [17]. Within our study 10% of youth said they were not yet ready to start ART, highlighting some of the challenges with linkage to care.

Few studies, mostly focused on adults, have investigated the feasibility and outcomes of ART initiation in the community [35–37]. Two studies investigated the feasibility of initiating ART in the community either following an HIV self-test in Malawi [36] or in the context of door-to-door HIV testing in Lesotho [37] with follow-up care provided at the clinic. ART initiation increased in both studies and viral suppression (<100 copies per ml) 12 months post diagnosis was higher (50.4%) in the community ART initiation group compared to the standard of care group (34.3%) in Lesotho. Barnabas *et al* investigated the effect of community- compared to clinic-based ART delivery (i.e. initiation and ongoing care) on viral suppression among adults newly diagnosed with HIV in Uganda and South Africa [35]. Participants enrolled in the trial were diagnosed at clinics, through HIV testing at community locations and at home including distribution of HIV self-test kits. Data on linkage to care were not available. However among those initiating ART viral suppression (<20 copies per ml) at 12 months was 74% among those receiving community-based ART compared to 63% among those receiving clinic-based ART. Reanalysing our data using the same cut-point for viral suppression (<20 copies per ml) and among those who did have an HIV viral load the prevalence of viral suppression in young people newly diagnosed with HIV was 78% (83/107). This is comparable with the results achieved in adults in Uganda and South Africa, but it does not take into consideration young people who have not been linked to care.

The CHIEDZA trial actively sought to limit the disruption of existing service provision. Thus young people previously diagnosed with HIV and in care with other service providers were encouraged to remain with their providers. This was largely achieved as only 6% of young people living with HIV transferring their care to CHIEDZA. However, CHIEDZA offered additional services for those known to be HIV-positive including community based HIV VL testing, adherence counselling if found to be unsuppressed and adherence and peer support through the CAPS groups. HIV VL testing at CHIEDZA was taken up by two thirds of eligible young people known to be HIV positive and in care elsewhere; more than three-quarters of them were virologically suppressed, comparable to population level estimates for this age group in Southern Africa [14, 38]. A minority of young people newly diagnosed or known

HIV positive registered with CAPS. Possible reasons for limited uptake of CAPS was the distance from young people's homes to the CHIEDZA centre, the interruptions caused by COVID19 lockdowns and prevailing transport restrictions due to the pandemic. These limitations are also likely to have affected general access to CHIEDZA.

We acknowledge several study limitations. This paper did not have a standard of care group and thus we could not compare outcomes of young people receiving community-based with those who received clinic-based ART. However, the viral suppression rates among those who were previously diagnosed and in care somewhere else were comparable with those receiving care at CHIEDZA. Also the small sample size and limited sociodemographic variables available did not allow a more in-depth analysis of risk factors for not linking to care. Linkage to care outside CHIEDZA was based on self-report which may be subject to social desirability bias and overestimate linkage particularly among youth who did not link to care at CHIEDZA.

## Conclusion

Our study provides evidence that provision of differentiated HIV services incorporating each step of the HIV care cascade for young people in the community is feasible. In an era of widely available HIV testing avenues, young people diagnosed with HIV through a community-based model may be harder to reach than those already diagnosed and continued attention to supporting linkage to and retention in care, particularly among youth, is needed.

## Supporting information

**S1 File. CHIEDZA trial protocol v4.0.**
(PDF)

**S1 Checklist/ TIDieR checklist.**
(PDF)

## Acknowledgments

The authors would like to acknowledge the CHIEDZA clients, communities and staff for their contribution to this study.

## Author Contributions

**Conceptualization:** Chido Dziva Chikwari, Katharina Kranzer, Mandikudza Tembo, Richard Hayes, Rashida Abbas Ferrand.

**Data curation:** Tsitsi Bandason.

**Formal analysis:** Katharina Kranzer, Victoria Simms, Amani Patel, Tsitsi Bandason.

**Funding acquisition:** Richard Hayes, Rashida Abbas Ferrand.

**Investigation:** Amani Patel, Constance Mackworth-Young, Sarah Bernays, Constancia Mavodza, Rashida Abbas Ferrand.

**Methodology:** Chido Dziva Chikwari, Katharina Kranzer, Victoria Simms, Amani Patel, Mandikudza Tembo, Owen Mugurungi, Richard Hayes, Constance Mackworth-Young, Sarah Bernays, Constancia Mavodza, Tsitsi Bandason, Ethel Dauya, Rashida Abbas Ferrand.

**Project administration:** Chido Dziva Chikwari, Mandikudza Tembo, Owen Mugurungi, Edwin Sibanda, Onismo Mufare, Lilian Ndlovu, Joice Muzangwa, Rumbidzayi Vundla, Abigail Chibaya, Constancia Mavodza, Fadzanayi Hove, Tsitsi Bandason, Ethel Dauya.

**Resources:** Owen Mugurungi, Edwin Sibanda, Rashida Abbas Ferrand.

**Supervision:** Chido Dziva Chikwari, Sarah Bernays, Ethel Dauya, Rashida Abbas Ferrand.

**Visualization:** Katharina Kranzer.

**Writing – original draft:** Chido Dziva Chikwari.

**Writing – review & editing:** Chido Dziva Chikwari, Katharina Kranzer, Victoria Simms, Amani Patel, Mandikudza Tembo, Owen Mugurungi, Edwin Sibanda, Onismo Mufare, Lilian Ndlovu, Joice Muzangwa, Rumbidzayi Vundla, Abigail Chibaya, Richard Hayes, Constance Mackworth-Young, Sarah Bernays, Constancia Mavodza, Fadzanayi Hove, Tsitsi Bandason, Ethel Dauya, Rashida Abbas Ferrand.

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
