## [Decision Letter · Decision Letter 0]

20 Nov 2023

PGPH-D-23-01727

Differentiated care for youth across the HIV care cascade in Zimbabwe

Dear Dr. Dziva Chikwari,

Thank you for submitting your manuscript to PLOS Global Public Health. After careful consideration, we feel that it has merit but does not fully meet PLOS Global Public Health’s publication criteria as it currently stands. Therefore, we invite you to submit a revised version of the manuscript that addresses the points raised during the review process.

We look forward to receiving your revised manuscript.

Kind regards,

Joel Msafiri Francis, MD, MS, PhD

Academic Editor

Journal Requirements:

1. We would like to request copy editing.

2. Please provide separate figure files in .tif or .eps format.

Additional Editor Comments (if provided):

Reviewers' comments:

Reviewer's Responses to Questions

**Comments to the Author**

1. Does this manuscript meet PLOS Global Public Health’s publication criteria? Is the manuscript technically sound, and do the data support the conclusions? The manuscript must describe methodologically and ethically rigorous research with conclusions that are appropriately drawn based on the data presented.

Reviewer #1: Yes

Reviewer #2: Yes

2. Has the statistical analysis been performed appropriately and rigorously?

Reviewer #1: I don't know

Reviewer #2: I don't know

3. Have the authors made all data underlying the findings in their manuscript fully available (please refer to the Data Availability Statement at the start of the manuscript PDF file)?

Reviewer #1: Yes

Reviewer #2: Yes

4. Is the manuscript presented in an intelligible fashion and written in standard English?

Reviewer #1: Yes

Reviewer #2: Yes

5. Review Comments to the Author

Reviewer #1: Summary of the study

This is a study nested within a cluster randomised trial that described a comprehensive community based HIV package for young people living with HIV in Zimbabwe called Chiedza. The package covered all aspects of the HIV cascade from HIV testing to linkage to care, ART initiation, viral suppression and retention. The authors concluded that the intervention had a positive impact on viral suppression, with 90% of young people living with HIV being virally suppressed (<1000c/ml) at six months. However, the authors reported that linkage to care still remained a challenge in this population with 30% not linking to care and 38% not being retained on ART. This is an important study as it has provided evidence for the short term effectiveness of a community-based HIV package for young people living with HIV that covered a wide geographic area. The major limitations of this study are the lack of comparison group and small sample size in the CHIEDZA intervention which may have been impacted by Covid 19 lock downs and travel restrictions.

Overall, this is a scientifically sound study but there may be some areas that require improvement.

Major Issues

1. The aims of this study have not been clearly articulated in the manuscript. In the final paragraph of the introduction paragraph (line 106-109), the authors state that they are describing the results of a differentiated models of service delivery offering HIV testing, linkage and ART initiation. From the methods section, it appears that viral suppression (<1000copies/ml) is the primary aim. However, the results section has reported other outcomes such as linkage to care and ART initiation (Figure 1 and table 1). Were these outcomes also considered as secondary objectives? To enhance clarity, it would be useful for the reader to have the aims clearly described in the introduction section of the manuscript.

2. Methods section

i)Outcomes section (line 193-195)- Clear definitions have been given for the outcomes: linked to care and unknown linkage. The outcome, not linked to care has not clearly defined. Can the authors provide a clear definition for clients who were categorised as not linking into care under this section.

ii)How was linkage to ART services with other service providers verified? Was it simply through self report by clients only (as described in line 191-192 in outcomes section) or were medical records checked as well? If these outcomes were obtained primarily through self report, this may introduce social desirability bias, and perhaps over-estimate the linkage to care in this group. This could be listed as a potential study limitation.

3. Statistical analysis section

This section is only describing how the outcomes were defined and how they were measured.

The data analysis has not been described in this section. The authors should consider including a clearly defined analysis plan describing the statistical techniques used and software used to conduct the analysis.

4. Results section

There were some inconsistencies with the calculation of some of the percentages as follows:

i)Line 273- 828/1156 is 71.6% not 76%. (Proportion of previously diagnosed youth aged 20-24)

ii) Line 275- CD4 count <200. The percentage is incorrect. 22/228 is 9.6% and not 20%.

iii) Line 283-284- The CAPS registration numbers for those accessing Chiedza and those accessing care elsewhere do not add up. Total enrolled into CAP is 323/1539 (21%). Of these 110 out of 323 who enrolled in CAPS were accessing care through Chiedza, so percentage should be 110/323 (34.1%) and not 110/300. Those enrolled into CAPS but accessing HIV care else where should be 213/323 (65.9%) instead of 213/111 (19%).

5. Additional analyses

The authors present results for viral suppression at a cut off of <20copies/ml in the results section - line 261-262. However, this cut off has not been described as one of the outcomes in the methods section. It is unclear why this viral load cut off is included in the results. Was this an exploratory analysis or part of the aims?

Minor issues

1. There needs to be consistency with the use of decimal places when reporting the percentages. Some of the percentages have been rounded off to whole numbers, others are reported with one decimal place in the text section of the results as well as in the table 1 and figure 1

2. To improve readability of some of the results , the authors may consider adding percentages to the following:

Line 273-274- Perhaps add the percentages for higher linkage to care in 20-24 year age group compared to 16-19 year age group.

Line 274- Add percentage for those with CD4 count available -228/234 (97.4%)

3. There is lack of clarity in two sentences in the results section.Please consider rephrasing

i) Line 244-245-is unclear. Please consider rephrasing as follows: Among those newly diagnosed, 265/377 (70.3%) decided to access care, 234/265 (88%) at CHIEDZA and 31/265 (12%) at other service providers

ii)Line 255-256- difficult sentence to read. Consider paraphrasing as follows: A minority of young people newly diagnosed with HIV and linked to care through CHIEDZA, 3.8% (9/234) and 19.4% (6/34) clients linked through other providers did not start ART at follow-up.

4. Covid 19 lock downs have been mentioned as a potential reason for low enrolments in the CAPS programme. Were enrolments into CHIEDZA also affected by the pandemic as well? This could be another study limitation.

5. Line 356-357 in the discussion section- It is not clear how the percentages for those who are virally suppressed (77.3%) has been calculated. Can you please provide the numerator and denominator for that. The previous viral load suppression proportions reported are 89% from figure 1.

,

Reviewer #2: Line 92: "young people are more likely than any other age group to disengage."

- this needs a reference

Line 95: "Importantly the bulk of attrition occurs before ART initiation and within the first 6 months on treatment.(12, 25)"

- Is this still the case, given rapid/same day ART?

Line 124 "Services included HIV testing, HIV treatment and adherence support"

- briefly describe adherence support

Line 129: "The intervention was specifically configured to be “youth friendly”

- How do you know? Were needs assessments done? Were youth included in the design?

Line 185: "Outcomes"

- can you think of a better title? If this section is meant to provide definitions (which is not clear) then disengagement should also be defined

Line 207 "Data management and statistical analysis"

- This section doesn't contain any information about statistical analysis. It seems percentages were calculated. Are there any other statistical analyses the authors think might be useful to do?

Line 331: "Few studies have investigated the feasibility and outcomes of ART initiation in the community and those did focused on adults only."

Reif L, Bertrand R, Rivera V, Jospeh B, Anglade B, Pape JW, et al. A novel model of community cohort

care for HIV-infected adolescents improves outcomes. Top Antivir Med. 2017; 25(1s):355s–6s.

Reif LK, Rivera VR, Bertrand R, Belizaire ME, Joseph JB, Louis B, et al. “FANMI”: a promising differentiated

model of HIV care for adolescents in Haiti. J Acquir Immune Defic Syndr. 2019; 82(1):e11–3.

347: "said they were not ready to start ART (n=11)"

- 10% (11/112) is a high figure and this deserves a brief comment in the Discussion

Line 390: "VS, TB and AP conducted data analysis"

- It took 3 people to calculate a few percentages?

6. PLOS authors have the option to publish the peer review history of their article (what does this mean?). If published, this will include your full peer review and any attached files.

**Do you want your identity to be public for this peer review?** For information about this choice, including consent withdrawal, please see our Privacy Policy.

Reviewer #1: No

Reviewer #2: No

---

## [Editor Report · Decision Letter 1]

19 Jan 2024

Differentiated care for youth in Zimbabwe: outcomes across the HIV care cascade

PGPH-D-23-01727R1

Dear Dr. Dziva Chikwari,

We are pleased to inform you that your manuscript 'Differentiated care for youth in Zimbabwe: outcomes across the HIV care cascade' has been provisionally accepted for publication in PLOS Global Public Health.

Best regards,

Joel Msafiri Francis, MD, MS, PhD

Academic Editor